# Adiponectin, Leptin, and Resistin Are Dysregulated in Patients Infected by SARS-CoV-2

**DOI:** 10.3390/ijms24021131

**Published:** 2023-01-06

**Authors:** Fabio Perrotta, Filippo Scialò, Marta Mallardo, Giuseppe Signoriello, Vito D’Agnano, Andrea Bianco, Aurora Daniele, Ersilia Nigro

**Affiliations:** 1Department of Translational Medical Sciences, Università della Campania “Luigi Vanvitelli”/Hospital Monaldi, Via L. Bianchi, 80131 Napoli, Italy; 2CEINGE-Biotecnologie Avanzate Scarl, Via G. Salvatore 486, 80145 Napoli, Italy; 3Dipartimento di Scienze e Tecnologie Ambientali, Biologiche, Farmaceutiche, Università della Campania “Luigi Vanvitelli”, Via A. Vivaldi, 81100 Caserta, Italy; 4Department of Mental Health, University of Campania “Luigi Vanvitelli”, 80138 Naples, Italy; 5Dipartimento di Medicina Molecolare e Biotecnologie Mediche, Università degli Studi di Napoli “Federico II”, Via Pansini, 80131 Napoli, Italy

**Keywords:** COVID-19, adipose tissue, HMW adiponectin, leptin, resistin, SARS-CoV-2

## Abstract

Obesity, through adipose tissue (AT) inflammation and dysregulation, represents a critical factor for COVID-19; here, we investigated whether serum levels of adiponectin, HMW oligomers, leptin, and resistin are modulated and/or correlated with clinical and biochemical parameters of severe COVID-19 patients. This study included 62 severe COVID-19 patients; 62 age and sex-matched healthy subjects were recruited as a control group. Anthropometric and biochemical parameters were obtained and compared. Adiponectin, HMW oligomers, leptin, and resistin were analyzed by ELISA. The adiponectin oligomerization state was visualized by Western blotting. When compared to healthy subjects, total adiponectin levels were statistically lower in severe COVID-19 while, in contrast, the levels of leptin and resistin were statistically higher. Interestingly, HMW adiponectin oligomers negatively correlated with leptin and were positively associated with LUS scores. Resistin showed a positive association with IL-6, IL-2R, and KL-6. Our data strongly support that adipose tissue might play a functional role in COVID-19. Although it needs to be confirmed in larger cohorts, adiponectin HMW oligomers might represent a laboratory resource to predict patient seriousness. Whether adipokines can be integrated as a potential additional tool in the evolving landscape of biomarkers for the COVID-19 disease is still a matter of debate. Other studies are needed to understand the molecular mechanisms behind adipokine’s involvement in COVID-19.

## 1. Introduction

The spread of the severe acute respiratory syndrome coronavirus 2 (SARS-CoV-2) reached pandemic proportions as declared by the World Health Organization (WHO) in March 2020, causing the Coronavirus disease 2019 (COVID-19). The symptoms of SARS-CoV-2 infection are highly heterogeneous including asymptomatic infection, mild symptoms such as fever, dry cough, and dyspnea, and in more serious cases, multilobar pneumonia and acute respiratory distress syndrome (ARDS) [1]. The heterogeneity of these clinical manifestations is strongly linked to general health conditions, vaccination state, age, and concomitant morbidities (particularly hypertension, obesity, and/or type 2 diabetes) of COVID-19 patients [2,3,4,5]. Obesity, through the dysregulation of the adipose tissue’s (AT) endocrine functions, may compromise the immune surveillance system, thus representing a critical risk factor for various infections, complications, and mortality [6,7]. Regarding COVID-19, obesity has been regarded as a risk factor for both its establishment and a higher rate of severe illness, hospitalization rate, admission to an intensive care unit (ICU), and mortality [8]. The relationships between obesity and COVID-19 can be explained by various pathways. Indeed, AT contains different cell types with significant intrinsic inflammatory properties; in particular, AT is constituted essentially of adipocytes but also contains monocytes, macrophages, and other cells that together participate in the regulation of inflammatory processes. Furthermore, adipocytes express receptors able to bind infectious components that, in turn, activate signal transduction cascades releasing pro-inflammatory cytokines, and acute-phase reactants determining an imbalance of adipokine production [9,10]. Visceral Adipose Tissue (VAT) inflammation can be considered a key event that contributes to triggering metabolic dysregulation and amplifying the systemic inflammatory characteristic of COVID-19 patients with metabolic disorders [11]. It is also well established that the imbalance of adipokine secretion plays a pivotal role in promoting both systemic metabolic dysfunctions and cardiovascular diseases, therefore representing a key factor in the pathophysiology of various disorders [12]. Among others, adiponectin is an adipokine with a pivotal role in the regulation of energy balance and insulin sensitivity. Dysregulation in the adiponectin profile is considered one of the major causes of metabolic disorders [13]. Adiponectin is secreted as three oligomeric complexes, including trimer (67 kDa), hexamer (140 kDa), and a high-molecular-weight (300 kDa) multimer through the interaction with three main receptors, AdipoR1, AdipoR2, and T-cadherin, which play an important role in insulin-sensitizing and have cardiovascular protective effects [14]. Leptin influences multiple endocrine functions and bone metabolism, and in addition to its key function of modulating energy homeostasis, leptin promotes inflammatory responses [15]. Resistin, named for its capacity to resist insulin action, is a rodent adipocyte-specific secretory factor while, in humans, the most important source of this molecule seems to be the macrophages. Resistin is related to abnormal metabolism promoting increased blood glucose, adipocyte proliferation, and obesity [16]. Contrasting results on the levels of resistin have been reported in murine and human obesity and type II diabetes, but it is considered a possible link between adiposity and insulin resistance. Furthermore, resistin appears to be a pivotal mediator of insulin resistance associated with inflammatory conditions.

Dysregulated levels of adipokines can also contribute to the inflammatory state involved in the severe complications that can occur during the course of COVID-19 [17,18]. Indeed, although still not conclusive, adiponectin levels have been found to be significantly lower in patients with COVID-19, while higher leptin levels have been linked to the cytokine storm during the infection [19]. To the best of our knowledge, few data are available about adiponectin, leptin, and resistin serum levels [18,20].

In this scenario, the aim of this study was to investigate the serum levels of adiponectin, its HMW oligomers, leptin, and resistin in severe COVID-19 patients, focusing on their correlation with clinical and biochemical patient parameters. 

## 2. Results

### 2.1. Anthropometrical and Clinical Data

Data from 62 hospitalized COVID-19 patients admitted during the second wave, between December 2020 to April 2021 when the SARS-CoV-2 alpha variant was predominantly widespread among the Italian population [21], were compared with those from 62 control subjects. The comparison between the main biochemical and anthropometrical parameters is reported in table in Section 4.2. The two groups statistically differ in BMI (*p* < 0.001), NLR *(p* < 0.001), glucose *(p* < 0.001), triglycerides *(p* < 0.001), ALT (*p* = 0.027), and GGT *(p* < 0.001), which were statistically higher in the COVID-19 group than controls, while hemoglobin (*p* = 0.041), albumin *(p* < 0.001), total cholesterol, HDL *(p* < 0.001), and LDL *(p* < 0.001) *(p* < 0.001) were statistically lower. 

Subsequently, COVID-19 patients were divided, according to the clinical outcomes, into two sub-groups: 47 were discharged and 15 were admitted to the Intensive Care Unit (ICU) needing oro-tracheal intubation (IOT) for invasive ventilation or died. Table 1 describes the comparison between the main clinical and biochemical parameters regarding the two COVID-19 sub-groups. The results indicated that death or IOT patients were statistically older (*p* = 0.047), had higher LUS scores (*p* = 0.012), and lower PaO_2_/FiO_2_ ratios (*p* = 0.035). 

### 2.2. Compared to Healthy Controls, Adiponectin, Leptin, and Resistin Are Differentially Expressed in Severe COVID-19 Patients

To verify the potential involvement of AT dysfunction in COVID-19 disease, we measured the serum expression of adiponectin, leptin, and resistin by ELISA assay. Total serum adiponectin levels were statistically lower in COVID-19 patients compared to healthy controls (15.11 ± 3.4 vs. 22.92 ± 9.6 μg/mL; *p* < 0.001) (Figure 1A).

The leptin levels were significantly higher in COVID-19 patients than in controls (16.94 ± 10 vs. 12.18 ± 7.2 ng/mL; *p* = 0.004) (Figure 1B). Finally, as shown in Figure 1C, serum resistin levels were statistically higher in COVID-19 patients compared to healthy controls (5859.79 ± 1953 vs. 4165.57 ± 1483 pg/mL; *p* < 0.001).

Successively, to investigate whether the adiponectin, leptin, and resistin levels could be considered a potential predictive indicator of COVID-19 disease, we performed a Receiver operating characteristic (ROC) curve analysis for the prediction of COVID-19 disease (Figure 2).

In detail, regarding adiponectin, when we compared controls to patients, we observed that, at an optimal cut-off point of 19.5 g/mL, corresponded a sensitivity of 62.9% and a specificity of 90.3% (AUC = 0.811); for leptin, an optimal cut-off point of 15.3 ng/mL corresponded to a sensitivity of 58.1% and a specificity of 71.8% (AUC = 0.638); for resistin, an optimal cut-off point of 4435.5 pg/mL corresponded a sensitivity of 73.68% and a specificity of 75% (AUC = 0.787). To further establish the influence of gender in adipokines’ expression and potential use as a predictive indicator of COVID-19, we also included a ROC curve for gender (Appendix A). Finally, we performed a multivariate analysis to discriminate the potential interference of other factors (Appendix A); after adjusting for BMI, glucose, cholesterol, and triglycerides, the levels of serum adiponectin and resistin were significantly associated with COVID-19 (*p* < 0.001 for both). To evaluate the prognostic factors of negative outcomes in severe COVID-19 patients, we analyzed biochemical and clinical parameters (Appendix A). Deceased or IOT COVID-19 patients had statistically increased levels of Dimer-D, IL-6, and KL-6 (*p* = 0.007, *p* = 0.048, *p* < 0.001, respectively). Adipokines do not differ between the two subgroups and therefore do not correlate with patient outcomes.

### 2.3. Oligomeric Distribution of Adiponectin in COVID-19 Patients 

Subsequently, considering that the complex regulation of adiponectin oligomeric status is involved in its distinct biological effects, we analyzed the distribution of serum adiponectin oligomers in COVID-19 patients in comparison to the control group by Western blotting (Figure 3).

Three bands corresponding to HMW (≥250 kDa), MMW (~180 kDa), and LMW (~70 kDa) oligomers were evident for both controls and COVID-19 patients (Figure 3). The densitometric evaluation of oligomeric distribution showed that COVID-19 patients had a lower expression of the three different oligomers compared to controls.

### 2.4. HMW Adiponectin Correlates with Leptin and LUS Score, Resistin Is Associated with IL-6, IL-2R, and KL-6

The analysis of the correlation between the adipokines demonstrated that HMW adiponectin was negatively correlated with leptin (Spearman’s rho −0.304; *p* = 0.017) (Table 2). No significant associations were found between adipocytokines and the main dysmetabolic serum parameters (including LDL and HDL cholesterol, triglycerides, and glucose).

Successively, we analyzed the association of adipokines with the biochemical and clinical parameters of COVID-19 patients. HMW adiponectin showed a positive association with LUS scores (Spearman’s rho 0.429; *p* = 0.004) (Table 2). 

Furthermore, analysis of the correlation between adipocytokines and inflammatory biomarkers showed the absence of significant association for adiponectin, HMW adiponectin, leptin, and their ratios. Conversely, resistin showed a weak positive association with IL-6, IL-2R, and KL-6 (*p* = 0.033, 0.014 and 0.037, respectively) (Table 3).

## 3. Discussion

The mechanisms underlying the severe predisposition that obesity represents for hospitalization, respiratory failure, and mortality in patients with COVID-19 are not yet fully understood but the poor outcomes observed in the obese population indirectly suggest the important role of AT in the hyper-inflammatory response during COVID-19 [22]. We analyzed the circulating levels of the three most relevant adipokines, i.e., adiponectin, its HMW oligomers, leptin, and resistin, in patients affected by the severe form of COVID-19 and our findings demonstrate a statistically different expression of the tested adipokines when compared to a healthy sex- and aged-matched population.

AT contains different cell types (adipocytes, monocytes, and macrophages) with significant intrinsic inflammatory properties that participate in the etiology of different pathologies [9]. AT crosstalk with peripheral tissues and organs is operated by the secretion of adipokines, hormones impacting different cellular processes such as energy metabolism and the inflammatory and immune responses against infections [23]. In light of this, dysregulated levels of circulating adipokines may play a role in the inflammatory state occurring in COVID-19 [18,24]. However, the literature on adipokines’ involvement in COVID-19 is yet contrasting. Among the adipokines, adiponectin and particularly its HMW oligomers have beneficial properties such as anti-inflammatory, antioxidative, and insulin-sensitizing effects and are reduced in obesity [25]. Here, we found that the levels of adiponectin in COVID-19 patients are statistically decreased and able to predict COVID-19 disease, as demonstrated by the ROC curve analysis. Interestingly, we did not find a bias determined by gender, suggesting that adiponectin regulation in COVID-19 is independent of sex. The usefulness of adiponectin as a marker of several other patho-physiological conditions of the lungs (such as COPD and cancer) has become clear in recent years [26,27,28,29,30]; on the other hand, the anti-inflammatory functions of adiponectin have been observed in different lung cell lines as well as in adiponectin-deficient mice that resulted in a predisposition to acute lung injury [31]. Though the functional role of adiponectin in COVID-19 is still unclear, a recent study found decreased adiponectin plasmatic levels in patients with acute respiratory failure associated with COVID-19 pneumonia when compared to non-COVID-19 patients [19]. These data seem to corroborate the results of our research. Interestingly, in our cohort of patients, HMW oligomers, recognized as the most active oligomeric form of adiponectin, are statistically decreased and correlated to LUS scores, implying a functional role of adiponectin in disease severity. Altogether, our evidence clearly indicates that adiponectin is functionally involved in COVID-19 pathogenesis. No published data are available regarding the HMW contribution to COVID-19, therefore additional studies are required to further explore such involvement.

The biological implication of leptin dysregulation in COVID-19 is not yet fully known. However, high circulating leptin levels have been hypothesized to be linked to the expansion of the cytokine storm related to COVID-19 aggravation, acute respiratory distress syndrome, and multiple organ failure, especially in obese patients [32,33]. In our study, we found that the leptin levels in the cohort of COVID-19 patients were higher than in controls and were possibly predictors of COVID-19, as demonstrated by the ROC curve analysis. Accordingly, hyperleptinemia has been described as one of the clinical characteristics associated with SARS-CoV-2 patients [34]. Van de Voort et al. also found that SARS-CoV-2 severe patients had higher levels of serum leptin compared to mild ones [34,35]. Notably, leptin levels have been associated with the immunologic abnormalities and systemic pro-inflammatory state seen in COVID-19 patients [34,35]. Lacobellis et al. suggested the use of leptin as a biomarker of inflammation useful to determine the risk of COVID-19 and its possible complications [36]. Here, although we found higher levels in patients, the value of this adipokine as a predictive biomarker was rather modest, and no correlation with clinical parameters was outlined. This partial discrepancy with published literature might be related to the heterogeneity of patients in different studies; indeed, it is somewhat difficult to standardize a patient’s conditions such as the degree of severity, medications, and comorbidities, making it necessary to further validate the usefulness of biomarkers in a larger cohort.

Resistin is an immunometabolic mediator that has been linked to numerous inflammatory disorders including, but not limited to, COPD, asthma, and cardiovascular diseases. In this study, we found that resistin levels are higher in COVID-19 patients compared to healthy controls and are able to predict COVID-19 disease, as demonstrated by the ROC curve analysis. The literature data available on resistin in COVID-19 showed, in accordance with our data, that resistin is higher in COVID-19 patients and predicts the requirement of invasive ventilation [20]. A very recent paper found resistin levels elevated in COVID-19 patients are also associated with cytokines and endothelial cell adhesion molecules and related to a worse outcome [37]. Similarly, we found that resistin is associated with IL-6, IL-2R, and KL-6.

In the more general context of lung diseases, resistin levels in cystic fibrosis (CF) patients have been correlated to disease status, identifying this molecule as a novel link between inflammation and lung disease in CF [38]. The main limitation of the present study is the low number of enrolled patients, which limits the statistical power of the analysis, confirming our findings in larger populations. However, to the best of our knowledge, there is only one published paper analyzing the three adipokines in COVID patients [18]. The authors conclude that leptin and adiponectin are associated with BMI, but not with clinical outcomes and inflammation of COVID-19 patients, while resistin, although not associated with BMI, is associated with worse clinical outcomes. Such data are partially in accordance with our data that demonstrate a negative correlation between leptin and HMW adiponectin but not with resistin; in addition, resistin correlates with some cytokines and HMW oligomers correlate with LUS scores. In different lung diseases, tuberculosis, interstitial lung disease, and asthma, adiponectin, leptin, and resistin have been analyzed together but no correlation among them was outlined [39,40,41].

In conclusion, our data strongly support the involvement of three adipokines in COVID-19, suggesting that the specific involvement of adipose tissue occurs in COVID-19. Further studies in more comprehensive cohorts are necessary to better elucidate the complex molecular mechanisms underlying adipokines’ involvement in COVID-19.

## 4. Materials and Methods

### 4.1. Subjects

A cohort of 62 patients (33 males, 29 females; age 68 ± 14.5) was admitted from December 2020 to April 2021 to the Vanvitelli Pneumonary Clinic at Monaldi Hospital in Napoli, Italy, for confirmed SARS-CoV-2 infection. At the time of hospital admission, the disease severity for each patient was assessed using the WHO COVID-19 severity categorization, and they had to have at least one positive real-time polymerase chain reaction (RT–PCR) test for SARS-CoV-2, signs of pneumonia on chest CT scan when hospitalized, and onset of symptoms no more than 3 days before hospital admission to be included in the analysis. Patients who died before the negative conversion of the viral swab test and those without laboratory tests performed within 48 h of admission to the hospital were excluded from the analysis.

The aliquots of serum samples were collected at hospital admission and stored at −80 °C for subsequent analysis. As controls, 62 age and sex-matched healthy volunteers were recruited from CEINGE—Biotecnologie Avanzate staff (Napoli, Italy). The study was conducted in accordance with the Declaration of Helsinki of the World Medical Association and was approved by the Ethics Committee of the Azienda Ospedaliera dei Colli, Napoli, Italy. The present research did not involve participants under the age of 18 years. Each patient gave fully informed consent for their participation in the study and the use of their biological samples for research purposes.

### 4.2. Anthropometric, Clinical and Biochemical Investigations

For each subject, age, weight, height, and BMI were registered. Clinical data and biochemical values are reported in Table 4. Arterial blood gas analysis was performed at patient admission to the ward. PaO_2_/FiO_2_ was calculated based on the ratio between arterial O_2_ pressure and oxygen inspiratory fraction administered. All patients underwent lung ultrasound evaluation and chest CT-Scan at admission to the hospital. LUS was performed adopting the 12-region model, 6 on each side, with each hemithorax divided into anterior, lateral, and posterior areas (delimited by the anatomical landmarks represented by axillary lines) and each area into upper and lower segments. Oxygen and ventilatory support to acute respiratory failure was provided through a venturi mask (VM), a non-rebreathing mask (NRM), a high-flow nasal cannula (HFNC), continuous positive airway pressure (CPAP), and/or pressure support non-invasive ventilation (NIV) according to available recommendations [42].

In addition, previous medical conditions such as chronic kidney failure (CKF), coronary heart disease (CHD), hypertension, type 2 diabetes, chronic obstructive pulmonary disease (COPD), anemia, thyroid disorders, obstructive sleep apnea syndrome (OSAS), asthma, and obesity were systematically collected.

### 4.3. Measurement of Adiponectin, Leptin, and Resistin

Total adiponectin serum concentrations were measured by enzyme-linked immunosorbent assay (ELISA) as previously reported [26]. A calibration curve was performed and quantified using human recombinant adiponectin (Biovendor R&D, Brno, Czech Republic) as a standard. The serum leptin and resistin amounts were detected using a commercial kit according to the manufacturer’s instructions (Invitrogen, Waltham, MA, USA). Each serum sample was assayed three times in duplicate.

### 4.4. Western Blot Analysis 

Sera from patients and controls were quantified for the total proteins by Bradford’s method (Bio-Rad, Hercules, CA, USA) and 10 µg of total proteins were treated with 1× Laemmli buffer, heated at 95 °C for 2 min, loaded on a 10% SDS-PAGE gel, and transferred as previously described [43]. The blots were scanned using the ChemiDoc MP imaging system (Bio-Rad, CA, USA) and analyzed via densitometry with ImageJ 1.52t software. Each sample was tested three times in duplicate.

### 4.5. Statistical Analysis

Categorical data were expressed as the number and percentage, while continuous variables were presented as both the median and interquartile range or mean and standard deviation, as appropriate according to the distribution. The main study variables were analyzed for possible associations with other clinical, biochemical, and respiratory variables using Spearman’s rank correlation coefficient. Multivariate linear regression analysis was performed with the stepwise method. The univariate analysis was performed using parametric (t Student for independent samples) and non-parametric statistics (U-Mann–Whitney) for continuous variables. The Fisher Exact test was performed for categorical variables. The potential association of adipokines in determining clinical outcomes in COVID-19 patients was assessed using a univariate model. Odds ratios (ORs) with 95% confidence intervals (CIs) were estimated by a logistic regression model with the significant variables of the univariate model and disease duration as covariates. Multivariate linear regression analysis was performed with the stepwise method. A *p*-value of <0.05 was considered statistically significant. All analyses were performed using statistical software STATA v16 (StataCorp. 2019. College Station, TX, USA: StataCorp LLC).

## Figures and Tables

**Figure 1 ijms-24-01131-f001:**
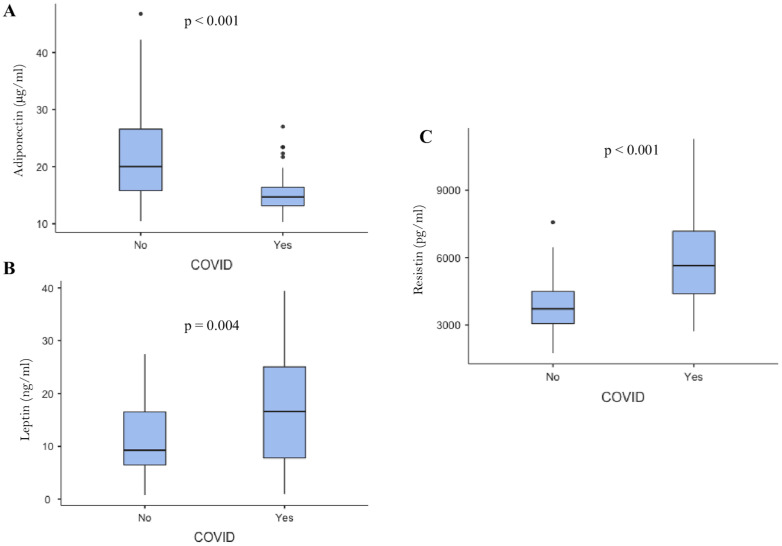
Serum total adiponectin, leptin, and resistin are differently expressed in severe COVID-19 patients compared to healthy controls. (**A**) Adiponectin levels were statistically reduced in patients diagnosed with severe COVID-19 compared to healthy controls (15.11 ± 3.4 vs. 22.92 ± 9.6 μg/mL; *p* < 0.001). (**B**) Leptin levels were significantly higher in COVID-19 patients than in controls (16.94 ± 10 vs. 12.18 ± 7.2 ng/mL; *p* = 0.007). (**C**) Resistin levels were significantly increased in patients diagnosed with COVID-19 compared to healthy controls (5859.79 ± 1953 vs. 4165.57 ± 1483 pg/mL; *p* < 0.0001).

**Figure 2 ijms-24-01131-f002:**
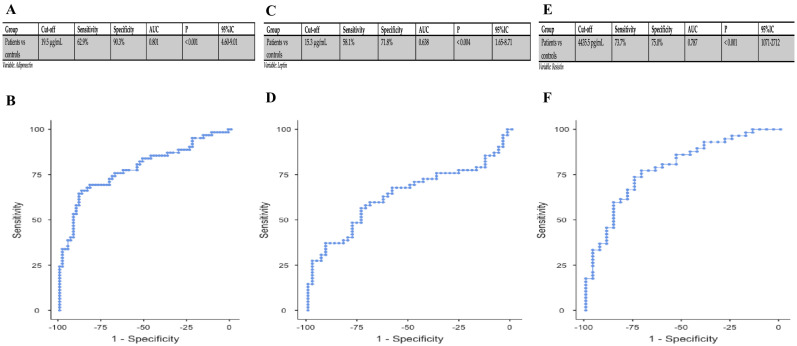
ROC curves for COVID-19 versus control of adiponectin (**A**,**B**), leptin (**C**,**D**) and resistin (**E**,**F**).

**Figure 3 ijms-24-01131-f003:**
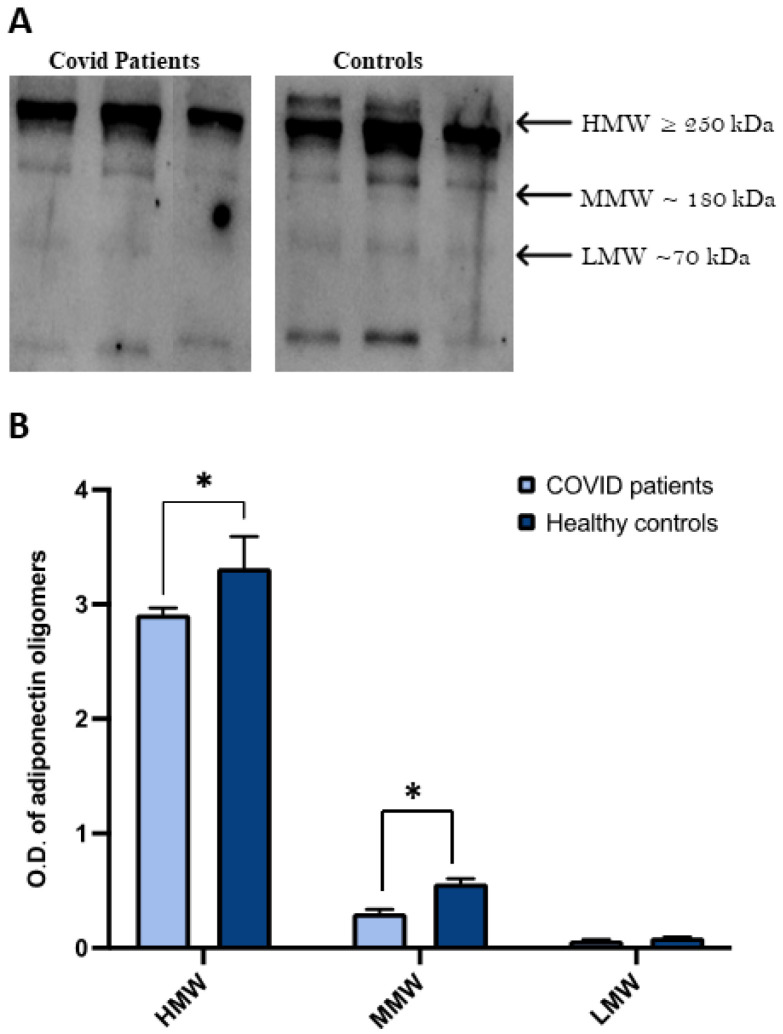
HMW oligomers in COVID-19 patients compared to controls. (**A**) Representative Western blot image for adiponectin oligomers, HMW, MMW, and LMW in the serum of controls and patients diagnosed with COVID-19. (**B**) Graphical representation of pixel quantization of analyzed controls and COVID-19 patients. * *p* < 0.05.

**Table 1 ijms-24-01131-t001:** Comparison of main anthropometrical and clinical parameters in discharged (n.47) vs. death or IOT (n.15) COVID-19 patients. Data are expressed as median value and interquartile range (IQR) or as absolute number and percentage.

	DISCHARGED (n.47)			DEATH or IOT (n.15)			TOTAL (n.62)			*p* Value
	Median	25th Percentile	75th Percentile	Median	25th Percentile	75th Percentile	Median	25th Percentile	75th Percentile	
**Gender**										0.373
*Male*	27(57.4%)			6 (40%)			33 (53.2%)			
**Age**	67	60	73	76	64.5	81.5	68	61	76.8	**0.047**
**BMI**	27.7	24.8	31.3	29.4	27.3	33.4	27.8	24.9	31.9	0.115
**Smoking**	8(17.0%)			2 (13.3%)			10 (16.1%)			1
**Comorbidities**										
*CKF*	1 (2.1%)			1 (6.7%)			2 (3.2%)			.
*CHD*	16 (34.0%)			7 (46.7%)			23 (37.1%)			.
*Hypertension*	30 (63.8%)			11(73.3%)			41(66.1%)			.
*Type 2 Diabetes*	10 (21.3%)			4 (26.7%)			14 (22.6%)			.
*COPD*	4 (8.5%)			2 (13.3%)			6 (9.7%)			.
*Anaemia*	4 (8.5%)			2 (13.3%)			6 (9.7%)			.
*Thyroid Disorders*	6 (12.8%)			2 (13.3%)			8 (12.9%)			.
*OSAS*	3 (6.4%)			1 (6.7%)			4 (6.4%)			.
*Asthma*	3 (6.4%)			0 (0.0%)			3 (4.8%)			.
*Obesity*	6 (12.8%)			4 (26.7%)			10 (16.1%)			.
**Respiratory Support**										
*Venturi or non-rebreathing mask*	4 (8.5%)			0 (0.0%)						.
*HFNC*	16 (34.0%)			1 (6.7%)						.
*CPAP*	22 (46.8%)			6 (40.0%)						.
*NIV*	4 (8.5%)			8 (53.3%)						.
**Treatment**										
*Corticosteroids*	44 (93.6%)			15 (100%)						.
*Anticoagulants*	43 (91.5%)			15 (100%)						.
*Remdesivir*	25 (53.2%)			5 (33.3%)						.
**LUS SCORE**	26	22.5	31.5	36	31.8	38	29	22.5	33	**0.012**
**CHUNG-SCORE**	13	11.5	14.5	13	12	16	13	12	15	0.22
**PaO_2_/FiO_2_**	120	88	171	93	84.5	107	105	85.8	140	**0.035**

Bold indicates statistically significant data.

**Table 2 ijms-24-01131-t002:** Correlation analysis between the adipokines and the LUS score.

		Adiponectin (μg/mL)	Leptin (ng/mL)	HMW (μg/mL)	Resistin (pg/mL)	LUS SCORE
Adiponectin (μg/mL)	Spearman rho	-				
	*p*-value	-				
Leptin (ng/mL)	Spearman rho	0.088	-			
	*p*-value	0.498	-			
HMW (μg/mL)	Spearman rho	0.011	**−0.304 ***	-		
	*p*-value	0.933	0.017	-		
Resistin (pg/mL)	Spearman rho	−0.073	**−0.270 ***	0.055	-	
	*p*-value	0.587	0.042	0.686	-	
LUS SCORE	Spearman rho	0.019	−0.036	**0.429 ****	−0.116	-
	*p*-value	0.904	0.820	0.004	0.477	-

Bold indicates statistically significant data. * *p* < 0.05, ** *p* < 0.01.

**Table 3 ijms-24-01131-t003:** Correlation between resistin and inflammatory markers.

		Resistin (pg/mL)	NLR	D-Dimer	Fibrinogen	CRP	LDH	IL2R	IL6	KL6
Resistin (pg/mL)	Spearman rho	-								
	*p*-value	-								
NLR	Spearman rho	0.146	-							
	*p*-value	0.284	-							
D-Dimer	Spearman rho	−0.137	0.166	-						
	*p*-value	0.319	0.210	-						
Fibrinogen	Spearman rho	0.139	**0.343 ***	0.024	-					
	*p*-value	0.427	0.033	0.885	-					
CRP	Spearman rho	0.165	0.124	−0.159	**0.565 *****	-				
	*p*-value	0.219	0.340	0.224	<0.001	-				
LDH	Spearman rho	0.062	0.164	**0.335 ***	0.069	0.070	-			
	*p*-value	0.649	0.214	0.010	0.677	0.595	-			
IL2R	Spearman rho	**0.296 ***	0.191	0.162	0.147	0.110	0.040	-		
	*p*-value	0.033	0.163	0.237	0.386	0.418	0.768	-		
IL6	Spearman rho	**0.338 ***	0.092	0.033	−0.001	0.116	**0.320 ***	0.166	-	
	*p*-value	0.014	0.503	0.813	0.997	0.396	0.016	0.222	-	
KL6	Spearman rho	**0.279 ***	0.239	0.072	0.013	−0.09	0.169	0.185	0.078	-
	*p*-value	0.037	0.065	0.590	0.940	0.468	0.201	0.173	0.566	-

Bold indicates statistically significant data. * *p* < 0.05, *** *p* < 0.001.

**Table 4 ijms-24-01131-t004:** Comparison of main biochemical parameters in COVID-19 patients vs. control group. Data are expressed as median value and interquartile range (IQR) or as absolute number and percentage. Bold indicates statical relevant data.

	Control Group			COVID-19 Patients			
	Median	25th Percentile	75th Percentile	Median	25th Percentile	75th Percentile	*p* Value
Gender: male (%)	26 (41.9)			33(53.2)			0.21
Age	61.5	54.3	75.3	68	61	76.8	0.092
BMI	25	24.1	25.7	27.8	24.9	31.9	**<0.001**
WBC	6.4	5.57	7.65	9.01	6.85	12.2	0.366
Neutrophils	3.6	2.33	4.45	7.98	5.61	10.7	**0.001**
Lymphocytes	2.06	1.58	2.52	0.73	0.57	1.14	**0.003**
Monocytes	0.362	0.277	0.43	0.53	0.32	0.72	0.355
Eosinophils	0.145	0.0918	0.239	0	0	0	**<0.001**
Basophils	0.0295	0	0.0584	0.01	0.01	0.02	**0.023**
NLR	1.74	1.27	2.12	11.2	5.9	15.5	**<0.001**
RBC	4.83	4.41	5.09	4.7	4.5	5.15	0.734
Haemoglobin	14.3	13	15	13.4	12.3	14.3	**0.041**
PLT	200	176	241	232	183	295	**0.01**
Glucose	89	79	98	140	104	203	**<0.001**
Creatinine	1	0.8	1.1	0.75	0.6	0.9	**<0.001**
Na+	141	140	142	138	135	140	**0.003**
K+	4.6	4.35	5	4.1	3.7	4.6	**<0.001**
AST	19	16	22	35	26	52.5	0.112
ALT	17	12	22	32	21.8	70	**0.027**
LDH	355	286	387	341	289	494	0.066
GGT	19	13	24	49	30	88	**<0.001**
CK	101	74	173	85	49.5	236	0.453
Alkaline phosphatase	63	54	78	55.5	47.8	76	0.493
Amylases	71	60.3	78	67.5	58.3	105	0.379
Bilirubin	0.53	0.37	0.68	0.68	0.51	0.91	**0.009**
Albumin	4.6	4.3	4.75	4.15	3.38	4.3	**<0.001**
Tot Chol	186	165	215	152	138	183	**<0.001**
HDL Chol	51.5	48	58.8	35.4	30.1	40.7	**<0.001**
LDL Chol	119	107	138	89.1	67.4	107	**<0.001**
Triglycerides	105	77	128	149	121	189	**<0.001**

## Data Availability

Not applicable.

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
