# Peer review of "Adiponectin, Leptin, and Resistin Are Dysregulated in Patients Infected by SARS-CoV-2"

_ijms, 2023, doi:10.3390/ijms24021131_

Round 1
Reviewer 1 Report
In this article, the author provides an interpretation of the potential role of adipose tissue dysregulation and some related proteins such adiponectin as a biomarker to predict Covid-19 severity.
All the work is based on the analysis of the main biochemical and anthropometrical parameters of 62 patients divided into 2 groups and on a finer analysis of the levels of the 3 potential biomarkers (adiponectin, resistin and leptin) via ELISA and WB.
Transversely to all the work, I believe moderate editing of the manuscript is necessary as some of the grammatical structures used are not always clear.
The data analysis as well as the approaches and conclusions are presented satisfactorily. The use of bibliographic references is appropriate. An objective analysis of the criticisms in the conclusions of this work has been appreciated.
In general, the topic introduced by this work is intriguing and could represent a good advance in the field.
Being the subject treated as interesting and able to represent a good advancement in the field, a more in-depth analysis of the issue would have been appreciated but, in general, I believe this work represents only the starting point for the study and identification of reliable biomarkers for the diagnosis of severe COVID-19.
Here are some notes and typos in the manuscript:
Line 49. "of patients with covid-19" is a repetition
Line 56. VAT: Please express the full name before using the acronym (as reported in the guidelines)
Line 63. upper case for Adiponectin
The sentence between line 64 and line 67 is not unclear as well as the following sentence.
Line 122. CHD: express the full name before using the acronym
Line 123. OSAS: express the full name before using the acronym
I generally suggest the author to check all acronyms used in the text and tables. Writing them in full the first time would help a lot the reader.
Line 167. It is not clear what the groups are and how they are divided.
Line 171. This sentence is not clear
Fig 2: both the blots are not straight and not aligned. I suggest the author to modify the figure.
Fig 2: I suggest the author to use "healthy" / “covid” instead of yes/no for the samples nomenclature.
Author Response
REVIEWER 1
Comments and Suggestions for Authors
In this article, the author provides an interpretation of the potential role of adipose tissue dysregulation and some related proteins such adiponectin as a biomarker to predict Covid-19 severity. All the work is based on the analysis of the main biochemical and anthropometrical parameters of 62 patients divided into 2 groups and on a finer analysis of the levels of the 3 potential biomarkers (adiponectin, resistin and leptin) via ELISA and WB. Transversely to all the work, I believe moderate editing of the manuscript is necessary as some of the grammatical structures used are not always clear. The data analysis as well as the approaches and conclusions are presented satisfactorily. The use of bibliographic references is appropriate. An objective analysis of the criticisms in the conclusions of this work has been appreciated. In general, the topic introduced by this work is intriguing and could represent a good advance in the field. Being the subject treated as interesting and able to represent a good advancement in the field, a more in-depth analysis of the issue would have been appreciated but, in general, I believe this work represents only the starting point for the study and identification of reliable biomarkers for the diagnosis of severe COVID-19.
We thank the reviewer for his/her precious comments. We do believe that the topic of our manuscript and our data might represent a step forward in the identification of reliable biomarkers for the diagnosis of severe COVID-19. We modified the text according to the reviewer’s comments and performed additional analysis to expand the usefulness of adiponectin and resistin as biomarkers. We also commented on this data in the revised version of the manuscript.
Here are some notes and typos in the manuscript:
Line 49. "of patients with covid-19" is a repetition
Response: This has been removed
Line 56. VAT: Please express the full name before using the acronym (as reported in the guidelines)
Response: This has been corrected
Line 63. upper case for Adiponectin
Response: This has been corrected
The sentence between line 64 and line 67 is not unclear as well as the following sentence.
Response: This has been corrected
Line 122. CHD: express the full name before using the acronym
Response: This has been corrected
Line 123. OSAS: express the full name before using the acronym
Response: This has been corrected
I generally suggest the author to check all acronyms used in the text and tables. Writing them in full the first time would help a lot the reader.
Response: This has been corrected
Line 167. It is not clear what the groups are and how they are divided.
Response: Here we have divided them as 47 discharged and 15 admitted in ICU
Line 171. This sentence is not clear
Response: we thank the reviewer and corrected the mistake
Fig 2: both the blots are not straight and not aligned. I suggest the author to modify the figure. (ERSILIA)
Response: we thank the reviewer and accordingly modified the figure
Fig 2: I suggest the author to use "healthy" / “covid” instead of yes/no for the samples nomenclature.
Response: This has been corrected

Reviewer 2 Report
Dear Authors,
The manuscript presented to me for review concerns the determination of the concentration of selected adipokines in the blood serum of patients with severe COVID-19, compared to healthy patients without COVID-19.
In my opinion, the work is not suitable for publication in this form due to the fact that the results obtained by the authors i.e. higher concentrations of leptin, resistin and low concentration of adiponectin in the group of COVID-19 patients may result not from the SARS-CoV-2 infection itself but from different amount and distribution of visceral or subcutaneous fat in these patients.
The authors used only the BMI to describe the patients. The electrical impedance to determine the amount of visceral or subcutaneous fat was not calculated. Additionally, patients with COVID-19 were in overweight (median BMI - 27.8) and compared to the control group with normal weight (median BMI -25); p <0.01!
The source of leptin and resistin can be visceral or subcutaneous fat and adiponectin is always lower in lean patients. There is no scientific proves to conclude that ongoing SARS-CoV-2 (even in severe infection) contributed to an increase in the serum concentration of selected adipokines in the overweight patients especially when compared to healthy volunteers whose BMI is normal.
The authors erroneously claim that higher levels of leptin and resistin are due to SARS-CoV-2 infection. In my opinion, this may be due to a greater amount of visceral or/and subcutaneous fat in the group of people with COVID-19 (BMI-27.8) vs healthy patients (BMI-25); p <0.01.
Furthermore, the authors did not test the content of adipose tissue by electrical impedance (BIA) in patients, but used only the anthropometric parameter i.e. BMI, a better solution seems to be the use of other anthropometric parameters as waist circumference (WC) and WHR (waist to hip ratio) for description and characteristics of the patients in particular in the context of fat-dependent concentration studies (adipokines).
It would also be wise to distinguish between men and women whose body fat distribution and content are completely different.
Moreover, there is no reason to conclude that adiponectin concentration may be an effective predictive indicator in the diagnosis of COVID-19 or the severe course of the disease in these patients. These differences in the concentrations of selected adipokines may be due to the different distribution and content of visceral or subcutaneous fat in these patients.
Author Response
REVIEWER 2
Dear Authors,
The manuscript presented to me for review concerns the determination of the concentration of selected adipokines in the blood serum of patients with severe COVID-19, compared to healthy patients without COVID-19. In my opinion, the work is not suitable for publication in this form due to the fact that the results obtained by the authors i.e. higher concentrations of leptin, resistin and low concentration of adiponectin in the group of COVID-19 patients may result not from the SARS-CoV-2 infection itself but from different amount and distribution of visceral or subcutaneous fat in these patients.
The authors used only the BMI to describe the patients. The electrical impedance to determine the amount of visceral or subcutaneous fat was not calculated. Additionally, patients with COVID-19 were in overweight (median BMI - 27.8) and compared to the control group with normal weight (median BMI -25); p <0.01!
The source of leptin and resistin can be visceral or subcutaneous fat and adiponectin is always lower in lean patients. There is no scientific proves to conclude that ongoing SARS-CoV-2 (even in severe infection) contributed to an increase in the serum concentration of selected adipokines in the overweight patients especially when compared to healthy volunteers whose BMI is normal.
The authors erroneously claim that higher levels of leptin and resistin are due to SARS-CoV-2 infection. In my opinion, this may be due to a greater amount of visceral or/and subcutaneous fat in the group of people with COVID-19 (BMI-27.8) vs healthy patients (BMI-25); p <0.01.
Furthermore, the authors did not test the content of adipose tissue by electrical impedance (BIA) in patients, but used only the anthropometric parameter i.e. BMI, a better solution seems to be the use of other anthropometric parameters as waist circumference (WC) and WHR (waist to hip ratio) for description and characteristics of the patients in particular in the context of fat-dependent concentration studies (adipokines).
It would also be wise to distinguish between men and women whose body fat distribution and content are completely different.
Moreover, there is no reason to conclude that adiponectin concentration may be an effective predictive indicator in the diagnosis of COVID-19 or the severe course of the disease in these patients. These differences in the concentrations of selected adipokines may be due to the different distribution and content of visceral or subcutaneous fat in these patients.
Authors Response
We really thank the referee for the extremely important considerations raised and we acknowledge that some points need to be addressed; in particular:
- When we started recruiting patients for adipokines testing we hypothesized that the expression of these hormones in COVID-19 patients could be different with possible implications on justifying the impact of obesity in COVID-19. Therefore, a significant BMI difference between Severe COVID 19 patients and control group was pre-planned. To understand the influence of BMI in Adipokines modulation and how this affects the statistical significance, we have conducted a multivariate analysis (See Supplemetary Tables). In the final multivariate model, adiponectin and resistin were statistically associated to COVID-19 regardless of the correction for other variables (BMI included)
- The possibility of BIA measurements is absolutely pertinent and would result in important information for the readers. However, the potential contamination of the BIA analyser along with the clinical scenario of patients with COVID-19 pneumonia (e.g. concomitant use of respiratory support, concurrent monitoring machineries) were considered as factors potentially limiting the feasibility of this test. This point has been addressed in the discussion section as potential study limitation.
- In the revised version of Manuscript, gender subgroup analysis has been performed and included in result section (We also included a ROC curve for each gender).
We are confident that in the actual version of the manuscript the points raised have been significantly addressed. Whenever some aspects have been discussed as potential study limitations.
We believe that this work will lead to much greater understanding of the role of adipose tissue in modulating the course of SARS-Cov2 infection leading to identification of reliable disease biomarkers and we thanks again for the quality of the your revision.

Reviewer 3 Report
Perrotta et al investigate levels of 3 different adipokines during COVID-19.
Although not novel, data reinforce current literature and present new insights into adiponectin forms.
I think however results are confused in some paragraphs, and the manuscript would benefit from a more comprehensive data-analysis
1-In table 1, it seems there is confusion with presented data: LDL, HDL is lower in the covid-19 patients, while the contrary is written in the text.
2-In table S1, LDH is statistically significant but the p-value is not in bold
3-It is not clear why authors calculated ROC only for adiponectin and not leptin or resistin. Similarly, why model coefficients were not considered for Leptin?
4-The use of Adiponectin for COVID diagnosis is, in my opinion, very controversial for several reasons:
Many measures are statistically different between healthy and covid patients. Some of them are even easy to measure than adiponectin. Why thus proposing adiponectin?
As suggested by authors in the discussion, adiponectin levels could be modulated in other diseases. Thus, lower adiponectin levels could be a very non-specific sign of inflammation/pathology.
5-To improve the novelty of the findings, authors should try to unravel more the role of those parameters usually associated with altered AT. E.g. Are the 3 adipokines correlated among each other? And with LDL, TG and HLD levels? And with IL-6?
The paper is of interesting but several aspects have been already covered by other manuscripts (e.g. https://doi.org/10.1111/cob.12568). So sentenced such as “the literature on adipokines involvement in COVID-19 is yet quite scarce” (discussion) should be removed. Similar sentences in the introduction should be also tuned down.
Author Response
Reviewer 4
Perrotta et al investigate levels of 3 different adipokines during COVID-19. Although not novel, data reinforce current literature and present new insights into adiponectin forms.
I think however results are confused in some paragraphs, and the manuscript would benefit from a more comprehensive data-analysis
- We sincerely thank the reviewer for his/her observations and suggestions. The aim of the manuscript was to reinforce the current literature data – as underlined by the reviewer – and also to make some progress on the state of the art by exploring the utility of testing adipokines in COVID-19. We revised the manuscript and added novel analysis according to the referee suggestions. We believe that the integration of these data has led to significant improvement in the quality of the manuscript, and we hope you may consider for publication
1-In table 1, it seems there is confusion with presented data: LDL, HDL is lower in the covid-19 patients, while the contrary is written in the text.
- We apologize for the mistake and accordingly modified.
2-In table S1, LDH is statistically significant but the p-value is not in bold
- We apologize for the mistake and accordingly modified.
3-It is not clear why authors calculated ROC only for adiponectin and not leptin or resistin. Similarly, why model coefficients were not considered for Leptin?
- We thank the reviewer for the suggestion. In the original version of the manuscript, we did perform the ROC curve for all the tested adypocytokines but we decided to present only Adiponectin because of the best performance in terms of AUC. However, we do acknowledge that could be of interest to the readers evaluate the ROC curves also for leptin and resistin; therefore, we have changed ROC Figure accordingly presenting all the data. Many thanks for this valuable comment.
4-The use of Adiponectin for COVID diagnosis is, in my opinion, very controversial for several reasons:
Many measures are statistically different between healthy and covid patients. Some of them are even easy to measure than adiponectin. Why thus proposing adiponectin?
As suggested by authors in the discussion, adiponectin levels could be modulated in other diseases. Thus, lower adiponectin levels could be a very non-specific sign of inflammation/pathology.
· We do agree with reviewer. The aim of this work was not to evaluate the relevance of adipocytokines in COVID-19 diagnosis but more properly to investigate the potential role of adipose tissue in COVID-19. Accordingly, we presented ROC curve to provide which median value of each adipocytokine is able to differentiate between healthy and COVID-19 subjects. Such analyses could have been a starting point to ideally understand if the adipokines are regulated and how in COVID-19, with the attempt to identify biomarkers or a combination of them. We agree with this reviewer and focus mainly on adiponectin that showed the highest difference between patients and controls. In the revised version of the manuscript, however, we included additional analyses focusing also on leptin and resistin. We provide ROC curve analyses (see new figure 2). Also, we do understand that in some part of the original manuscript, the role of adypocytokines was associated with COVID-19 diagnosis and in the revised version of the paper we have removed any sentence which may cause potential misleading interpretation from the readers.
5-To improve the novelty of the findings, authors should try to unravel more the role of those parameters usually associated with altered AT. E.g. Are the 3 adipokines correlated among each other? And with LDL, TG and HLD levels? And with IL-6?
- We thank the reviewer for his/her useful observations. Our initial aim was to unravel the modulation that occurs in adipokines during COVID-19 as a reflection of the involvement of adipose tissue in the disease. Therefore, we selected three adipokines and proceeded to the evaluation of their serum levels. As underlined by the reviewer, the manuscript lacks molecular aspects behind the regulation of the adipokines; therefore, in the revised version of the manuscript, we have performed additional analyses to investigate potential relationships among adipokines and clinical/biochemical parameters.
- We found that Leptin negatively correlates with HMW adiponectin (Spearman's rho -0.304; p = 0.017). No significant associations between adipocytokines and the main dysmetabolic serum parameters (including LDL and HDL cholesterol, triglycerides and glucose).HMW adiponectin has a positive association with LUS score (Spearman's rho 0.429; p=0.004). Furthermore, analysis of correlation between adipocytokines and inflammatory biomarkers showed absence of significant association for adiponectin, HMW adiponectin, leptin and their ratios. Conversely, Resistin showed positively weak association with IL-6, IL-2R and KL-6 (p=0.033, 0.014 and 0.037, respectively). We have added these data in the revised manuscript. Once again, many thanks for this valuable comment.
The paper is of interesting but several aspects have been already covered by other manuscripts (e.g. https://doi.org/10.1111/cob.12568). So sentenced such as “the literature on adipokines involvement in COVID-19 is yet quite scarce” (discussion) should be removed. Similar sentences in the introduction should be also tuned down
- We agree with the reviewer and accordingly modified the text. We intended to underlie that the analysis of a combination of more than two adipokines have not been considered before (the publication which you are referring to had not yet been published at the time of writing our paper). We added the novel published data in the reference list. Also, thanks to the reviewer’s suggestions, in the revised version of the manuscript, we have added substantial data that we hope have improved the quality and novelty of the manuscript.
Round 2
Reviewer 2 Report
Dear Authors/Editor
I maintain my position regarding the non-approval of the manuscript submitted to me for publication.
In my opinion, the work is not suitable for publication in this form due to the fact that the results obtained by the authors i.e. higher concentrations of leptin, resistin and low concentration of adiponectin in the group of COVID-19 patients may result not from the SARS-CoV-2 infection itself but from different amount and distribution of visceral or subcutaneous fat in these patients (measured with BMI) as well as different level of glucose in both groups (p<0.001)!
Selecting subjects for the control group with a BMI statistically significantly lower especially in the context of the fat-related adipokines cannot be the basis for conclusions proposed by the authors.
Moreover, leptin is an adipokine secreted from visceral adipose tissue and its level is proved to be higher in patients with T2DM. Lack of information about T2DM and insulin resistance (IR) in COVID-19 group as well as healthy donors ? Authors provide this informations only in in discharged vs death or IOT COVID-19 patients (Table. 2|). Results presented in Figure 1 are related with severe COVID-19 patients compared to healthy controls.
Without this informations we cannot made a clearer conclusions, especially that COVID patients group have higher level of glucose (median 140 mg/dl) vs healthy volunteers (median 89 mg/dl); there is no information on the number of patients with DMT2 in both groups (control- healthy and COVID-19 patients).
Leptin, adiponectin and other fat related adipokines are related to glucose metabolism. Authors demonstrated higher level of leptin in COVID group (patients with median glucose level 140 mg/dl) when compared to healthly control (median glucose level 89 mg/dl). In humans, plasma adiponectin levels were correlated negatively with adiposity, insulin resistance, type 2 diabetes , and metabolic syndrome , yet positively correlated with markers of insulin sensitivity. Prospective and longitudinal studies indicated that lower adiponectin levels were associated with a higher incidence of type 2 diabetes Moreover, Insulin stimulates both leptin biosynthesis and secretion from adipose tissue establishing a classic endocrine adipo-insular feedback loop; the so-called “adipo-insular axis” In this case, hiperinsulinemia along with higher glucose level may be responsible for leptin over expression in COVID-19 group (authors did not rule out this possibility).
Author Response

(The authors gave the same response as above.)

Reviewer 3 Report
Authors have done properly improved the manuscript.
Author Response
Many thanks for your kind review of the manuscript.